# Enhancing Pullulan Production in *Aureobasidium pullulans* through UV Mutagenesis Breeding and High-Throughput Screening System

**Shuyue Zhang [1,2], Zhe Feng [1,2], Qingming Zeng [3], Junhao Zeng [3], Huijing Liu [1,2], Pan Deng [1,2], Shangyu Li [1,2], Nan Li [1,2] and Junqing Wang [1,2,*]**

[1] State Key Laboratory of Biobased Material and Green Papermaking (LBMP), Qilu University of Technology (Shandong Academy of Sciences), Jinan 250353, China; 17861406507@163.com (S.Z.); woshifengzhe888@163.com (Z.F.); 19862126880@163.com (H.L.); 19374619909@163.com (P.D.); 10431230766@stu.qlu.edu.cn (S.L.); linan1166@163.com (N.L.)

[2] School of Biological Engineering, Qilu University of Technology (Shandong Academy of Sciences), Jinan 250353, China

[3] Shandong Mimei Biological Technology Co., Ltd., Weifang 262600, China; 17864099688@163.com (Q.Z.); zhuohanw0717@163.com (J.Z.)

* Correspondence: wjqtt.6082@163.com

**Abstract:** This study addresses the production enhancement of pullulan, an extracellular polysaccharide with various applications. Pullulan is primarily produced by *Aureobasidium pullulans* (*A. pullulans*), and genetic modification is commonly used to increase its yield. However, there is a need for a more efficient and safer method. To achieve this, we designed a high-throughput screening system utilizing a unique fluorescent protein specific to pullulan. Ultraviolet (UV) mutagenesis was applied to create a pool of mutant strains, and flow cytometry allowed for single-cell screening. Our approach yielded strain M1-B3, which exhibited a substantial increase in pullulan production from 26.5 g/L to 76.88 g/L. Additionally, the molecular weight of the produced pullulan significantly increased, expanding its potential commercial application. This study demonstrates an efficient and safe method to enhance pullulan production in *A. pullulans*. The UV mutagenesis and flow cytometry based on screening not only increased yield but also improved pullulan's molecular weight. The adaptability of this method to other polysaccharides and its potential for genomic analysis and broader applications make it a valuable tool in bioproduction.

**Keywords:** pullulan; *Aureobasidium pullulans*; bimolecular fluorescence complementation (BIFC); UV mutagenesis; mutant strains; flow cytometry

## 1. Introduction

Pullulan is an extracellular polysaccharide first discovered in *Aureobasidium pullulans* (*A. pullulans*) by Bauer [1]. Its structure consists of repeating units of maltotriose linked by α-1,4-glycosidic bonds, and the units are connected by α-1,6-glycosidic bonds [2]. The relative molecular weight ranges from approximately $4.5 \times 10^4$ to $6 \times 10^5$ Da. The viscosity of pullulan solution is directly proportional to the molecular weight of pullulan, with its molecular weight being significantly influenced by various parameters [3]. Pullulan, a non-toxic, harmless, transparent, oil- and oxygen-blocking water-soluble polysaccharide, has been used for various purposes, such as plasma substitutes, food products, adhesives, cosmetic additives, and flocculants, offering considerable commercial value. The solution of pullulan also has a stable pH range; even if burned, it does not release any harmful gases and can be spontaneously decomposed by microorganisms even if discarded [4].

*A. pullulans* is a yeast-like fungus commonly found in nature, which has been studied for almost 100 years, exhibiting polymorphic characteristics that adapt to changes in the growth environment. It is widely distributed and is found in many places, including fresh

water, seawater, soil, plant leaves and flowers, rock surfaces, and artificial surfaces [5]. It manifests five distinct cell shapes, including yeast-like cells, chlamydospores, blastospores, swollen blastospores, and mycelium. *A. pullulans* is also called "black yeast" because it also produces melanin in the later stages of fermentation. Melanin produced by *A. pullulans* also has many physiological functions; for example, it has antioxidant and free radical scavenging activities, as well as anti-tumor, antiviral, anti-radiation, liver damage protection, anti-aging, and other biological activities. Melanin is the main by-product of *A. pullulans,* and pullulan is the main product of *A. pullulans* [3].

To enhance pullulan production, most researchers increase the yield through genetic modification. Wang et al. found that after knocking out the *CreA* gene in the strain, the yield of pullulan increased by approximately 10%, and the molecular weight also decreased [6]. Conversely, Liu et al. constructed a strain devoid of melanin production by knocking out the alb1 gene, resulting in a 41.01% production reduction. Intriguingly, by adding melanin to the culture medium, the production of pullulan increased by 31.44% compared to the original data [7]. Some researchers choose to improve the yield of pullulan by optimizing the fermentation conditions. Wu et al. discovered that the yield of pullulan was affected by temperature, so they designed an experiment with two stages of fermentation temperature, which increased the yield by about 27.8% [8]. Sheng et al. found that uracil could enhance the accumulation of pullulan; after adding uracil, the yield of pullulan increased by 11.35 g/L [9]. It was also reported by Kumar et al. that a 10:1 carbon/nitrogen ratio is the most favorable condition for pullulan production [10]. In many studies, the ability of *A. pullulans* to produce pullulan was improved by knocking out gene or other methods, which may sometimes have some limitations and uncertainties.

This study introduces a high-throughput screening breeding system incorporating a bimolecular fluorescent protein that is tailored to detect pullulan. Bimolecular fluorescence complementation (BIFC) refers to using two proteins with interaction affinity to bring the fluorescent protein fragments connected to them closer and assemble them into a complete fluorescent protein, thereby characterizing the occurrence and spatial location of protein interactions [11]. When combined with high-throughput screening, this system enabled the selection of high-yielding strains following UV mutagenesis ultimately enhancing the ability of *A. pullulans* to produce pullulan. The application of this screening system can not only improve the screening efficiency but also screen the strains with high yield, which is of importance to meet the demand of the polysaccharide market.

## 2. Materials and Methods

### 2.1. Strains and Media

The strain utilized in this experiment, *Aureobasidium pullulans* M1 (ATCC 201428), was obtained through screening by our laboratory. *E. coli* BL21 (DE3) was designed and purchased from Shanghai Bioengineering (St. Louis, MO, USA). Standard pullulan was sourced from Sigma Aldrich (St. Louis, MO, USA). All chemicals used were of analytical grade and obtained from commercial suppliers. The culture medium for E. Coli consisted of LB, comprising 0.5 g/L Yeast extract, 1 g/L Tryptone, 1 g/L NaCl, 0.1 g/L amplification, with a pH of 7 (solid medium supplemented with 2% agar). *A. pullulans* was cultivated in a medium composed of 2 g/L Tryptone, 2 g/L Yeast extract, 50 g/L sucrose, 1 g/L NaCl, 0.6 g/L $(NH_4)_2SO_4$, 6.3 g/L $K_2HPO_4$, and 0.2 g/L $MgSO_4$ (solid medium supplemented with 2% agar).

### 2.2. Construction of Plasmid

The bimolecular fluorescent protein BIFC-CBM41-FC was used to construct a fluorescent detection label for pullulan [12]. The bimolecular fluorescent protein can bind pullulan polysaccharide specifically and thus play the role of labeling. In this experiment, two fluorescent proteins a and b were connected to BIFC-CBM41. The two proteins with interaction affinity bring their connected fluorescent protein fragments closer together and assemble into a complete fluorescent protein, thereby generating fluorescence as the filter mark. In this study, the N-terminus ($A_{502}$) of the fluorescent protein BIFC-CBM41 was linked to

protein a ($A_1 \sim C_{465}$) through the linker ($A_{466} \sim C_{501}$) to form fusion protein 1-a. Similarly, the C-terminus ($A_{309}$) of the gene BIFC-CBM41 was linked to protein b ($A_{346} \sim G_{594}$) through the linker ($A_{310} \sim C_{345}$) to form fusion protein 1-b. These synthetic *1-a* and *1-b* gene fragments were synthesized by Shanghai Sangon and inserted into the pET-20b (+) vector at the 5′ restriction site *Nde* I and the 3′ restriction site *Xho* I, resulting in the creation of recombinant plasmids pET-20b (+)-1-a-BIFC-CBM41 and pET-20b (+)-BIFC-CBM41-1-b. *E. coli* BL21 (DE3) competent cells were prepared, and the two recombinant plasmids were transformed into these cells after enzyme digestion. Positive clones were selected to obtain the *E. coli* engineering strains a1 and b1 expressing fusion proteins 1-a and 1-b. (The plasmid maps are shown in Supplementary Figure S2.)

### 2.3. Expression and Purification of Pullulan Screening Marker

To assess the expression effects of fluorescent proteins, strains a1 and b1, each containing "his" tag, were inoculated into a 50 mL ampicillin-resistant LB liquid medium with an initial inoculation volume of 0.4%, and the culture was maintained at 37 °C. Induction was initiated by adding 0.5 mmol/L IPTG when the $OD_{600}$ was between 0.6 and 0.8, and incubation continued at 25 °C for 8–12 h. Subsequently, cells were harvested by centrifugation at 12,000 r/min for 10 min, followed by sonication (250 W 30 min) in a phosphate buffer (0.2 M, pH 7.4) for 20 min. The resulting homogenate was then centrifuged at 12,000 r/min for 15 min at 4 °C to remove cell debris. The protein supernatant was added to a pre-equilibrated Ni-NTA Sepharose column, which was subsequently washed with an impurity wash buffer (20 mM Tris, 250 mM NaCl, 20 mM imidazole, pH 7.4). Bound proteins were eluted using elution buffer (20 mM Tris, 250 mM NaCl, 200 mM imidazole, pH 7.4). Protein concentration was determined using the Bradford method (also called Coomassie blue staining, which is a method for determining protein concentration), and protein purity was analyzed through SDS-PAGE analysis [13]. (The images of Western blots are shown in Supplementary Figure S1.)

### 2.4. Induced Expression of Fluorescent Detection Proteins and Optimization of the System

The protein supernatant of 1-a and 1-b were previously purified via the Ni column as the fluorescent label [14]. Fluorescence signals were detected using a microplate reader (excitation light at 501 nm, emission light at 527 nm). The experiment was divided into four parts: (1) Added the protein supernatants 1-a and 1-b and the 3% pullulan in the 200 μL system. In the control groups, 1 × PBS, 3% xylan, and 3% dextran were added in place of the 3% pullulan. (2) Added 3% pullulan in the 200 μL system, 1 × PBS, protein supernatant 1-a, protein supernatant 1-b, protein supernatant 1-a and 1-b were four variables. They were added to the system, respectively. (3) Three different ratios were detected: 2:2:1, 2:1:1, and 1:2:1, comprising fluorescent protein supernatants 1-a and 1-b, along with the 3% pullulan solution. (4) Fluorescence intensity was monitored at hourly intervals over a 6 h period. The optimal ratio, marked by the strongest fluorescence, and the optimal time reaction, indicated the largest change in fluorescence, were determined through this assessment.

### 2.5. Construction of Mutant Strain Library through UV Mutagenesis

*A. pullulans* was activated at 28 °C on a solid medium, with a single colony subsequently selected and inoculated into the seed medium [15]. The culture was maintained at 28 °C for 48 h. When the $OD_{600}$ value reached 1.2~1.4, 2 mL of *A. pullulans* liquid was extracted and centrifuged to remove the supernatant, and then, an equal volume of sterilized phosphate buffer was added. The mixture was homogenized and evenly spread onto a disposable plate. Prior to irradiation, the UV lamp was pre-heated, and the disposable plate was positioned 20–30 cm from the lamp. Starting at 0 s, samples were collected at 20 s intervals during irradiation. The sample selected at 0 s was used as a control. Subsequently, the irradiated solution was spread onto separate plates for different periods of time and incubated overnight at 28 °C. The fatality rate was calculated based on the number of colonies [16].

The irradiation time at which the mortality reached 90% (which can achieve a high positive mutation rate) was designated as the standard time for ultraviolet mutagenesis [17]. The mutated mold liquid was then inoculated into the seed culture medium and cultured in the dark at 28 °C for 48 h. Afterward, 20 mL of the seed culture liquid was centrifuged at 10,000 r/min for 10 min. The supernatant was removed, and an equal volume of phosphate buffer was added and mixed evenly. The *A. pullulan* liquid was filtered using rapid filter paper, which had been pre-sterilized with UV in an ultra-clean station. The filtered *A. pullulan* liquid was then determined according to the optimal ratio. It was combined with a fluorescent protein and allowed to react for the optimal duration. *A. pullulan* strains were subsequently screened using flow cytometry based on fluorescence intensity [18,19].

### 2.6. Flow Cytometry Screening of Mutant Strains

The flow cytometer in this experiment used 0.85% high-purity NaCl as the sheath fluid and mixed fluorescent microspheres (Beckman Coulter, Brea, CA, USA) to control the flow and optical system. Samples were prepared and subsequently analyzed and sorted using a Moflo XDP model ultrafast flow cytometer (Beckman Coulter, Brea, CA, USA) equipped with a three-color laser. The fluorescent protein used for *A. pullulans* in this study exhibited its maximum excitation wavelength near 488 nm; thus, the 488 nm laser was used for optimal fluorescence detection. The fluorescent protein emitted light primarily at approximately 527 nm. For data collection, the FL1 and FSC channels of the experimental model flow cytometer were selected to capture the emission wavelength. Injector pressure differentials were adjusted to maintain a detection rate of approximately 1000 target particles per second, with 1/10,000 of the detected cells sorted into 96-well plates. The collected data were subsequently analyzed and processed using Kaluza, the dedicated data analysis software (version number: A82959) for the Moflo XDP flow cytometer (Beckman Coulter, Pasadena, CA, USA) [20,21].

### 2.7. Cultivation in Microplates and Fermentation after Screening

After culturing in a 96-well plate for 48 h at 28 °C and 200 rpm, we observed the bottom of the well plate. When we noticed cell clumps precipitating, we transferred 20 μL of the culture to a deep 96-well plate with liquid medium and continued the cultivation at 28 °C and 200 rpm for an additional 48 h. Subsequently, we pipetted 200 μL of fermentation broth from the deep 96-well plate and centrifuged for 10 min at 10,000 r/min, then mixed the supernatant with the fluorescent protein and measured the fluorescence intensity of the conjugate hourly. We selected the *A. pullulan* liquid with a more pronounced change in fluorescence intensity and transferred 200 μL into a 100 mL shake flask containing fermentation medium. This was maintained at 28 °C and 200 rpm for 48 h.

The selected strains were cultured in a 5 L fermentation tank; the volume of the medium in the 5 L tank is 3 L, following the sequence of solid medium activation, seed medium, and finally, the 5 L fermentation tank. The seed inoculation amount at each level was 5%, and the seeds were cultured until the $OD_{600}$ reached a range from 1.2 to 1.4 before inoculating the next level. These cultures were maintained at 28 °C, with a stirring speed of 500 r/min, a ventilation volume of 1 vvm, and a fermentation cycle of 84 h. Feeding was performed as needed, with samples taken every 12 h to monitor product changes.

### 2.8. Determination of the Yield and Viscosity of Strain Production "Pullulan"

We pipetted 10 mL of fermentation broth and centrifuged it for 10 min at 10,000 r/min. The supernatant was retrieved and mixed with two volumes of 95% ethanol while stirring, facilitating the precipitation of the polysaccharide. The resultant precipitate was thoroughly rinsed with deionized water to remove excess impurities. Subsequently, the precipitate was dried at 80 °C and weighed to obtain the approximate amount of pullulan extracted from the fermentation broth.

Another 20 mL sample of the fermentation broth was centrifuged at 10,000 r/min for 10 min. The supernatant was collected, and its viscosity (dL/g) was measured using

a capillary viscometer. Viscosity was determined by the product of the constant of the viscometer and the measured time(s) [22].

$$\text{Viscosity} = \text{the constant of the viscometer} \times \text{time (s)}$$

### 2.9. Determination of the Molecular Weight of Strain Production "Pullulan"

The dried pullulan was ground into a powder and prepared as a 1 mg/mL solution using pure water. The molecular weight of the polysaccharide produced from the selected mutant strain was then measured using high-performance liquid chromatography (HPLC). The molecular weight of the sample can be calculated according to its peak time.

This experiment used the SHIMADZU high-performance liquid chromatograph, with a TSKgel GMPWXL chromatographic column and detection facilitated by a differential refractive index detector (RID-20A). A mobile phase of 0.1 mol/L $NaNO_3$ was used at a flow rate of 0.5 mL/min and a column temperature of 30 °C. Injection volume was 10 µL, and four pullulan standards (Sigma, Darmstadt, Germany) with molecular weights ranging from 23 kDa to 894 kDa were selected as references [23]. The software included with the instrument was used to analyze the experimental data.

## 3. Results

### 3.1. Expression of Screening Marker and Optimization Results of Detection Conditions

This experiment revealed that simultaneous binding of fusion proteins 1-a and 1-b to adjacent sites on the substrate, pullulan, resulted in strong fluorescence production, while no significant fluorescence was produced when they did not bind to pullulan. Also, the two fluorescent proteins 1-a and 1-b do not fluoresce when they are present in solution with pullulan alone. Hence, fusion proteins 1-a and 1-b can serve as rapid detectors for pullulan within the system (Figure 1 shows a schematic diagram).

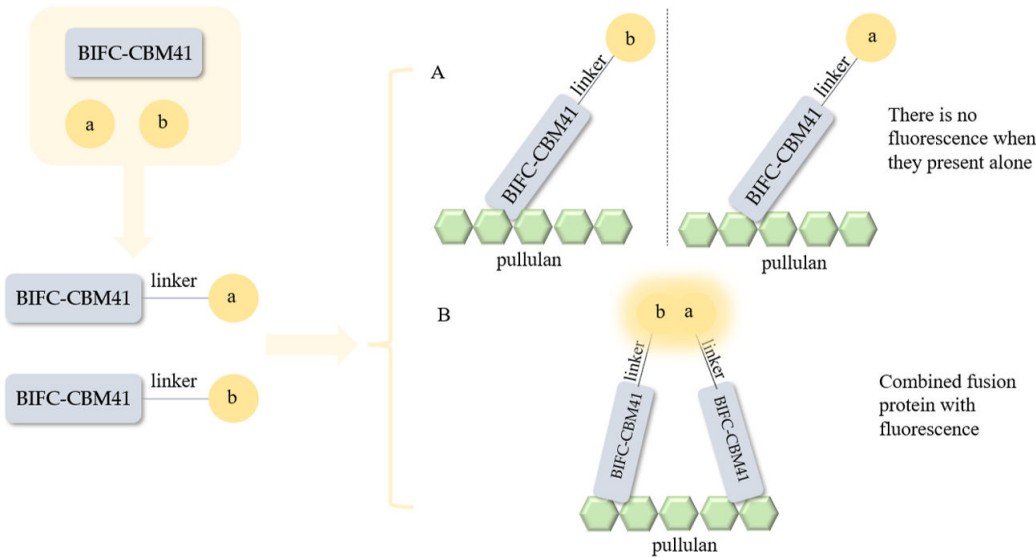

**Figure 1.** The schematic diagram of the BIFC. (**A**) shows the uncombined fusion proteins; (**B**) shows the combined fusion proteins.

Figure 2A illustrates that fusion proteins 1-a and 1-b can detect pullulan specifically. Compared with other samples, the fluorescence intensity of pullulan can reach 168,599 RFU (Relative Fluorescence Unit) ($p \leq 0.01$). Figure 2B shows that the stronger fluorescence intensity can be produced only when fusion proteins 1-a and 1-b are present in the system at the same time ($p \leq 0.001$), and neither will produce strong fluorescence when present alone. Figure 2C shows that the 2:2:1 ratio exhibited a better change in fluorescence intensity, surpassing the other two ratios by 12,000~28,000 RFU ($p \leq 0.01$). Figure 2D illustrates that after 1 h of fusion, the fluorescence intensity reached 166,908 RFU stabilizing within

three hours ($p \leq 0.001$). Consequently, only when there are both fusion proteins 1-a and 1-b and pullulan in the system can the system produce stronger fluorescence intensity. As a result, the 2:2:1 ratio and a 1-hour reaction time were identified as the optimal conditions for the fluorescence reaction system.

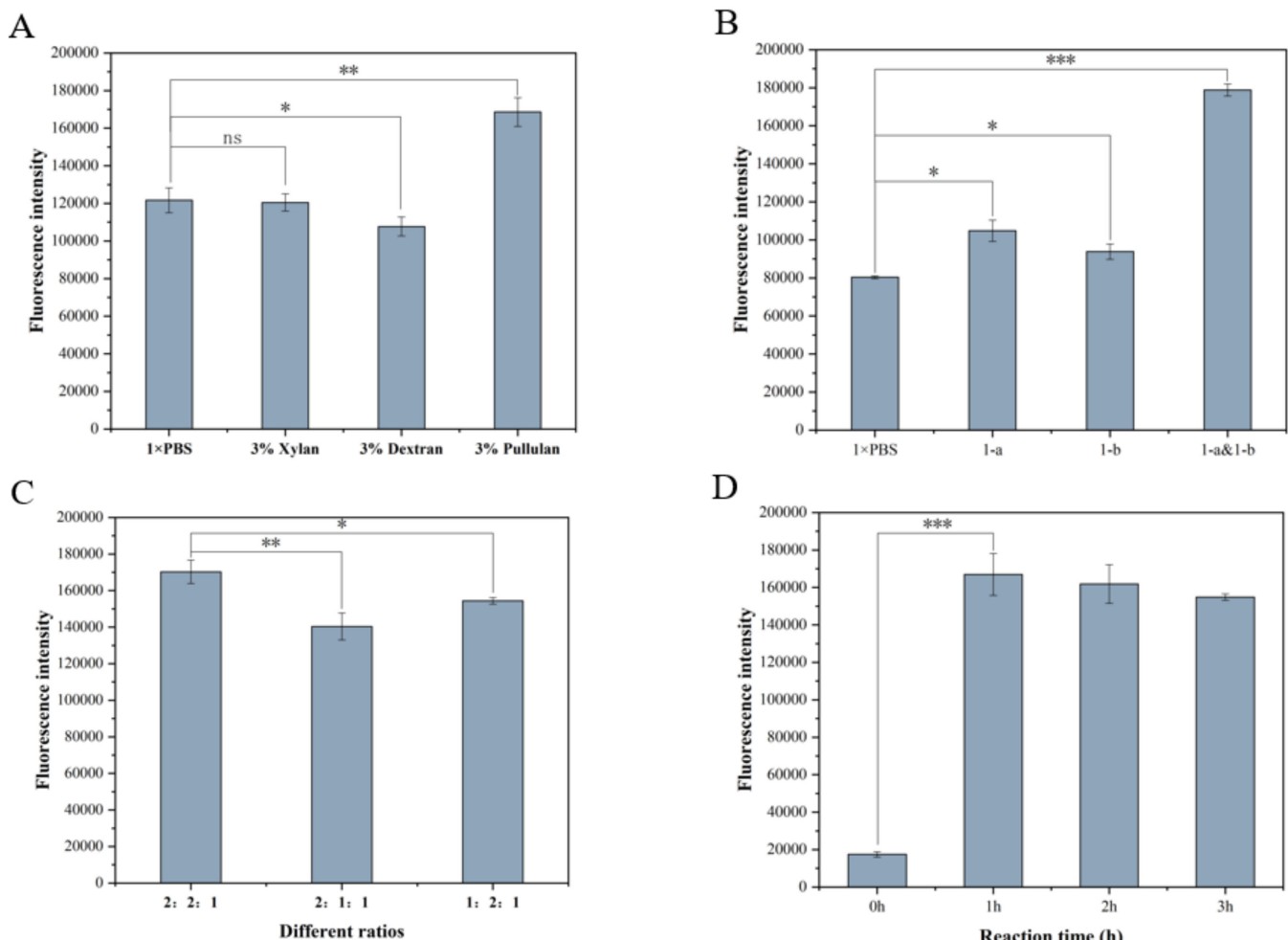

**Figure 2.** (**A**) shows the fluorescence intensity between $1 \times$ PBS, 3% xylan, 3% dextran, and 3% pullulan. (**B**) shows the fluorescence intensity with different compositions. (**C**) shows the proportion of different fluorescent proteins to the standard substance and the change in fluorescence intensity. (**D**) shows the change in fluorescence intensity at different times. ($p > 0.05$ (ns); $p \leq 0.05$ (*); $p \leq 0.01$ (**); $p \leq 0.001$ (***)).

### 3.2. Result of Mortality of A. pullulans Irradiated by UV

The mold liquid was exposed to fixed-point UV light for 20–120 s, with optimal UV treatment time parameters determined based on Figure 3. Notably, a mortality of 94.45–96.38% was achieved when UV irradiation lasted 80–100 s, leading to the selection of 90 s as the UV induction time. Further results from ultraviolet mutagenesis indicated that a mortality of 97.87% was achieved after 120 s of irradiation. The mortality of *A. pullulans* was observed to increase with prolonged irradiation time.

$$\text{Mortality} = \frac{\text{The number of colonies on plates without UV} - \text{The number of colonies on the plate after UV}}{\text{The number of colonies on plates without UV}} \times 100\%$$

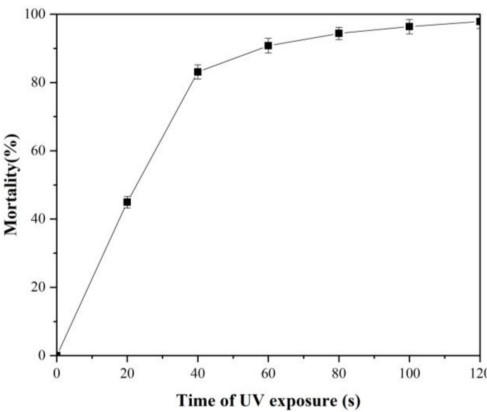

**Figure 3.** Relationship between time of ultraviolet mutagenesis and mortality. Took a sample every 20 s for coating experiments and calculated the mortality of *A. pullulans*.

### 3.3. Flow Cytometry Analysis and Sorting of Aureobasidium pullulans

Subsequently, the fluorescent protein was added to both mutated and unmutated strain mold liquids, followed by a one-hour incubation. Flow cytometry was used to detect and analyze the original strain, *A. pullulans* M1, and the mutant strain. As demonstrated in Figure 4A, the FITC value of the mutated strain predominately ranged from $10^5$ to $10^8$, whereas the unmutated strain displayed FITC values between $10^6$ and $10^7$. This marked a clear change in fluorescence in the mutated strain, the range of fluorescence has become wider. As shown in Figure 4A, the fluorescence of the mutated strain is distributed in the four regions: a, b, c, and d. Especially the strains in the d region are the target strains for our screening with high fluorescence.

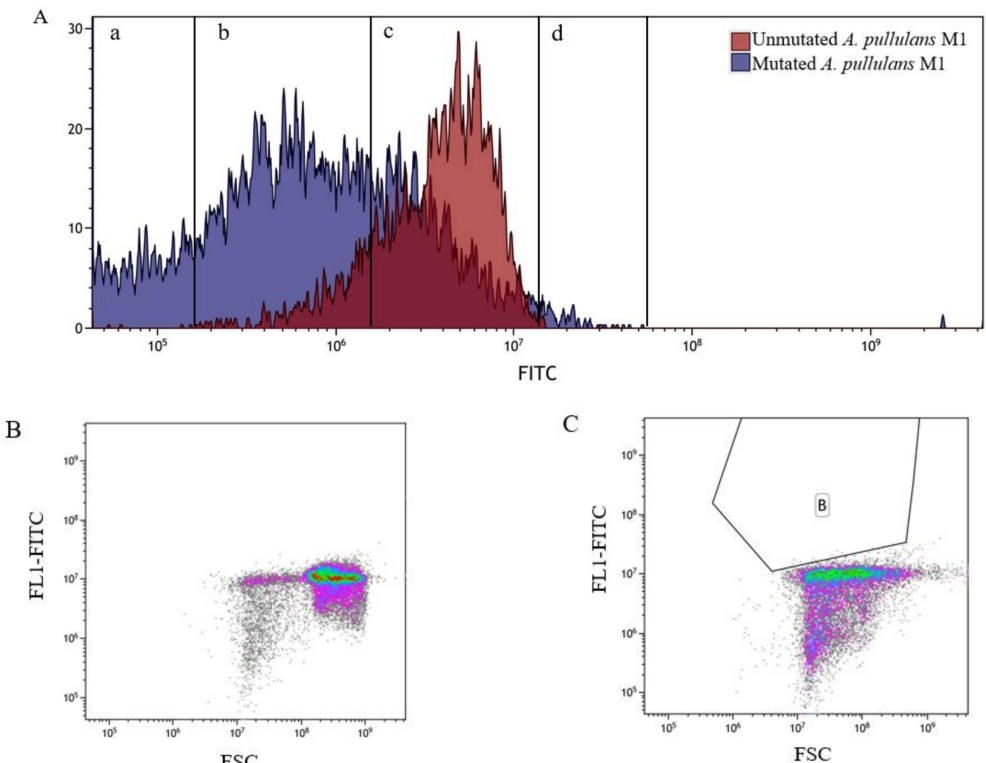

**Figure 4.** Microbial changes between mutated and unmutated strains by flow cytometry analysis. (**A**) Flow cytometry analysis of mutated *A. pullulans* and the unmutated. Regions a, b, c, and d represent different fluorescence intensity areas. (**B**,**C**) show the difference between unmutated and mutated strain M1 analyzed by flow cytometry. Gate B in (**C**) is the range of strains to be screened.

After a one-hour incubation, the analysis results in Figure 4 revealed a more dispersed pattern in the mutated strain compared to the unmutated ((B) was the result of the unmutated strain, and (C) was the result of the mutated strain). The reason for this result is that UV mutagenesis changed the characteristics of cells; some of the cells died, and some of them produced beneficial mutations, such as the cells with stronger fluorescence in the B gate. Cells in the B gate were selected as the mutated strain. The sorted areas were placed in a 96-well plate, with the unmutated strain serving as the control. A screening intensity of 1/10,000 (based on FSC, indicating cell size, and FITC, indicating the fluorescence intensity, dual parameter data, choose one cell for every 10,000 cells) was used, and sterile conditions were maintained during sorting to reduce the risk of target cell population contamination.

### 3.4. Screening of the Mutant Strains and Determination of the Product

Fluorescence intensity was measured, and three strains (M1-B3, M1-C5, and M1-H8) displaying stronger fluorescence (as shown in Figure 5) were selected for shake flask culture. The strain with the highest pullulan yield was chosen as the starting point for subsequent screening. This process was repeated to gradually increase the yield of the strain.

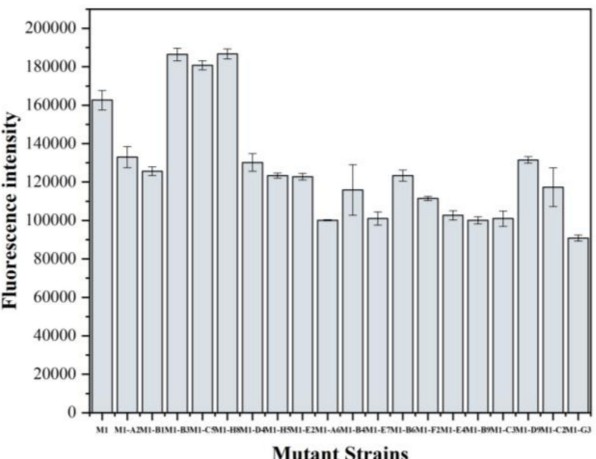

**Figure 5.** The change of fluorescence intensity of selected cells (include M1, M1-A2, M1-B1, M1-B3, M1-C5, M1-H8, M1-D4, M1-H5, M1-E2, M1-A6, M1-B4, M1-E7, M1-B6, M1-F2, M1-E4, M1-B9, M1-C3, M1-D9, M1-C2, M1-G3) and the fluorescence results of the reaction with BIFC after 1 h.

Figure 6A demonstrates that shake flask fermentation increased the pullulan yield from the initial 26.5 g/L to 41.59 g/L ($p \leq 0.001$). This indicates that after a round of screening, the ability of *A. pullulans* to produce pullulan was enhanced, especially the yield of strain M1-B3, which reached a yield of 41.59 g/L in the shake flask. Under identical conditions, strains M1-C5 and M1-H8 achieved yields of 31.33 g/L ($p \leq 0.01$) and 38.38 g/L ($p \leq 0.001$), respectively. Figure 6B illustrates that the viscosity of the pullulan solution corresponds to the weight-average molecular weight. The original pullulan produced by M1 had a molecular weight of approximately 209 kDa, while the screened strains exhibited molecular weights ranging from 390 to 899 kDa. This substantial improvement in molecular weight compared to M1 is evident. (The result of HPLC is shown in the Supplementary Figure S3.) The viscosity of pullulan produced by the selected strains also increased compared with M1. Compared with the original strain M1, these three parameters have all improved.

These results highlight that, in comparison to the original strain M1, the combined use of ultraviolet mutagenesis and fluorescence protein screening, followed by the cultivation of single cells with strong fluorescence, effectively identifies strains with higher yields, thereby increasing pullulan production.

A
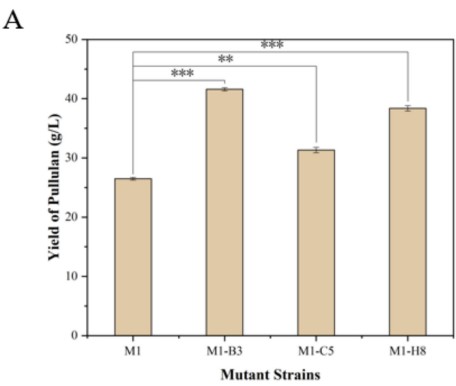

B
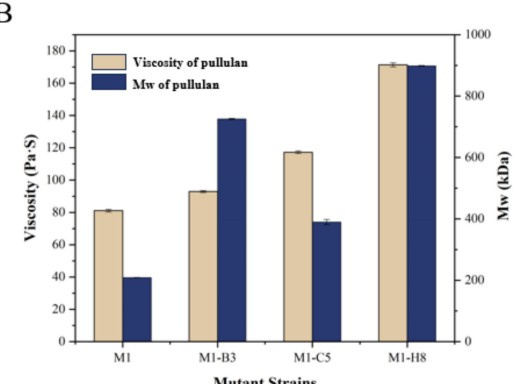

**Figure 6.** Detection of yield, viscosity, and molecular weight of fermentation products of strain M1, M1-B3, M1-C5, and M1-H8. (**A**) shows the yield of the strains. (**B**) shows the viscosity and molecular weight of the strains. ($p \leq 0.01$ (**); $p \leq 0.001$ (***)).

### 3.5. Fermentation of Mutant Strains

In the 5 L fermentation tank, samples were extracted every 12 h to measure the pullulan content in the fermentation broth. As depicted in Figure 7, the yield of the strain selected after mutation was significantly higher than the yield of the starting strain M1 in each time period, especially for M1-B3, which reached its highest yield of 76.88 g/L at 60 h ($p \leq 0.001$). M1-C5 and M1-H8 also reach the highest yield at 60 h. M1-B3 makes a remarkable increase of 41.95 g/L, which could be used in production.

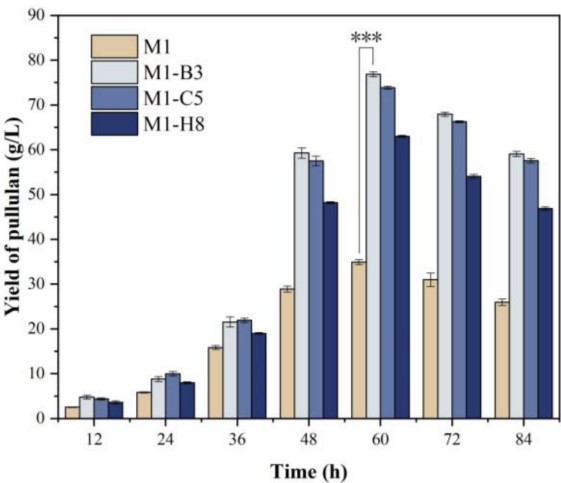

**Figure 7.** Detection of the yield of strains M1, M1-B3, M1-C5, and M1-H8 in 5 L fermenter every 12 h. ($p \leq 0.001$ (***)).

## 4. Discussion

In this study, we designed a fluorescence screening system that can specifically recognize pullulan by combining BIFC technology. BIFC analysis is characterized by high sensitivity because fluorescence recovery requires the interaction of two proteins, so fluorescence can only be restored when each protein is fused to the substrate [24]. Because of the intrinsic fluorescence of the complemented protein complex, BIFC does not require any special treatment of cells. Therefore, the interference with the normal cellular environment is almost zero [25]. The fluorescence screening system constructed in this experiment can specifically bind pullulan. It combines bimolecular fluorescent protein with pullulan at the same time and detects its fluorescence intensity as a screening standard to screen out cells with high yield.

Mutation breeding is a widely adopted method that accelerates breeding cycles, enhances efficiency, and yields strains with anticipated traits. Many studies have traditionally

employed chemical reagents such as colchicine and nitrosoguanidine to induce mutations. While their effect is significant, these chemical mutagens are often carcinogenic or highly toxic, resulting in low safety levels [26]. Ultraviolet (UV) radiation is a prevalent mutagen in higher plants, microorganisms, and algae due to its simplicity, which also has the advantages of convenience, safety, and a high mutagenesis rate, making it the ideal choice in this experiment [27]. This method effectively reduces the mutagenesis duration and increases the mutation rate. Ultraviolet mutagenesis generates a large number of mutant molds. Thus, the screening method plays a pivotal role. Kang et al. used a medium containing trypan blue for screening mutated strains [28], while Yu et al. combined ultraviolet mutagenesis with γ-ray and used colony color observation to select melanin-nonproducing strains [29]. Therefore, the ultraviolet irradiation used in this study not only ensured the safety of the experiment but also improved the efficiency of mutagenesis.

Flow cytometry (FCM) is a technology that uses flow cytometry for precise multi-parameter quantitative analysis and sorting of single cells or biological particles based on their physical–chemical properties [30]. In our prior research, flow cytometry was used to rapidly monitor complex microbial flora, enhancing winemaking quality [31]. However, there is currently no complete system for rapid detection and screening of pullulan through flow cytometry. Therefore, this study established an efficient and rapid screening system combining fluorescent proteins and flow cytometry [32]. This method can not only detect pullulan polysaccharides quickly and specifically but also provide an efficient system for screening high-yield strains.

In this study, the initial *A. pullulans* strain M1 was subjected to ultraviolet mutagenesis combined with fluorescent proteins, and the mutated strains were screened using flow cytometry technology. Screening relied on fluorescence intensity produced when the strain is combined with the fluorescent protein, resulting in the identification of strains with high pullulan production. While our study successfully demonstrated the efficacy of UV mutagenesis in enhancing pullulan production, it is essential to acknowledge certain limitations. Our research focused on *A. pullulans* and pullulan production, and the generalizability of the findings to other microorganisms or polysaccharides may vary. Additionally, further research and genome sequencing are needed to fully understand the genetic changes underpinning the improved pullulan production and to optimize this approach for practical applications. In the future, we will continue to study the relationship between the yield and molecular weight of pullulan cause the increased molecular weight of the pullulan indicates diverse potential applications, with higher molecular weights carrying greater commercial value [33,34]. Therefore, it is very promising to study the molecular weight of pullulan in subsequent research.

## 5. Conclusions

In our study, *A. pullulans* strain M1 was subjected to UV mutagenesis, combined with a fluorescent protein, and then followed by single-cell screening via flow cytometry and subsequent culture and yield measurements. Strain M1-B3 demonstrated a substantial increase in yield from 26.5 g/L to 76.88 g/L. This highlights the effectiveness of UV mutagenesis combined with fluorescent proteins in improving the ability of *A. pullulans* to produce pullulan. We successfully utilized ultraviolet mutagenesis and high-throughput screening with a bimolecular fluorescent protein to significantly increase the production of high-quality pullulan. This approach not only aligns with our objective of enhancing pullulan yield but also offers broader implications. Overall, our findings hold promise for both the bioproduction industry and the advancement of genetic research in microorganisms.

**Supplementary Materials:** The following supporting information can be downloaded at: https://www.mdpi.com/article/10.3390/fermentation10020103/s1, Figure S1: Protein electrophoresis of fluorescent proteins, strains a1(a) and b1(b). Figure S2: Plasmid map of fluorescent proteins, strains a1 and b1. Figure S3: The HPLC result of the standard pullulan samples and the production of the mutated strains in this study.

**Author Contributions:** J.W. and H.L. designed the experiments. S.Z. performed the investigation. Q.Z., J.Z., Z.F. and P.D. analyzed the results. S.L. and N.L. drafted the manuscript. All authors have read and agreed to the published version of the manuscript.

**Funding:** This work was supported by the Focus on Research and Development Plan in Shandong Province (2022CXGC020206, 2020CXGC010603, 2021ZDSYS10), Innovation Fund for Small and Medium-sized Technology Innovation Capacity Enhancement Project of Shandong Province (2023TSGC0765, 2023TS1047), Key innovation Project of Qilu University of Technology (Shandong Academy of Sciences) (2022JBZ01-06), Taishan Scholar Foundation of Shandong Province (tscx202306067) and National Natural Science Foundation of China (31801527).

**Institutional Review Board Statement:** Not applicable.

**Informed Consent Statement:** Not applicable.

**Data Availability Statement:** Data are contained within the article and Supplementary Materials.

**Acknowledgments:** We would like to thank the State Key Laboratory of Bio-based Materials and Green Papermaking, Qilu University of Technology, for its help and support and the Taishan industry-leading talent funding. We would also like to thank Ruiming Wang, Piwu Li, and Ting Wang who provided the funding.

**Conflicts of Interest:** Author Qingming Zeng and Junhao Zeng are employed by the company "Shandong Mimei Biological Technology Co., Ltd., China". But for purposes of this investigation, there was no financing relationship with the company; therefore, there are no conflicts of interest. The remaining authors declare that the research was conducted in the absence of any commercial or financial relationships that could be construed as a potential conflict of interest.

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
