# Peer review of "Enhancing Pullulan Production in Aureobasidium pullulans through UV Mutagenesis Breeding and High-Throughput Screening System"

_fermentation, doi:10.3390/fermentation10020103_

Round 1

Reviewer 1 Report

Comments and Suggestions for Authors

The manuscript "Enhancing Pullulan Production in A. pullulans Through High-Throughput Screening and Genetic Modification" presents interesting research results on the optimization of the synthesis of pullulan, an important metabolite of A. pullulans. Interesting genetic engineering methods were proposed that influenced the amount of pullulan synthesized, thanks to which the mutants could be used industrially. the manuscript is well planned, the results are correctly described, the figures are legible.

Detailed notes:

Chapter 2.4 - please refer to the literature.

line 112-113 - please describe the Bradford method.

Figure 6,7 - no statistical analysis of the results.

I suggest moving Figure S2 to Manuscript.

Reviewer 2 Report

Comments and Suggestions for Authors

The manuscript describes the development and application of a bimolecular high-throughput technique using fluorescence to screen pullulan-overproducing mutant strains of A. pullulan. The paper is quite interesting and well written, the results are presented in a methodical and scientific manner and the scientific techniques are adequately described.

I recommend the manuscript to be accepted for publication after minor revision.

Minor comments

Title: Please write in the title the full name of the microorganism, Aureobasidium pullulans

Line 22-23: In the abstract and in the main text (eg Line 333) you write that the increase in pullulan production was from 26.5 g/L (B1 strain) to 76.88 g/L. However, from your results (Figures 6 and 7) the pullulan production by strain M1 appears to be around 36 g/L. Please Check it out. In addition, it is best to give values with a precision of no more than their standard deviation.

Lines 212: Give the relative fluorescence intensity with standard deviations or write the mean value with as many significant figures as the precision of the measurement.

In addition, it is best to give values with a precision of no more than their standard deviation.

Comments on the Quality of English Language

no

Reviewer 3 Report

Comments and Suggestions for Authors

The manuscript entitled « Enhancing pullulan production in A. pullulans through high-throughput screening and genetic modification » contains two parts: UV mutagenesis of A. pullulans to obtain pullulan highly-producing mutant strains, and development of a high-throughput screening for pullulan detection based on BIFC. But these two parts do not clearly appear.

Overall, the presentation of this study is disjointed. The objectives are not clear, the manuscript is not properly organized, several pieces of information are missing.

There is no clear focus or rationale, although there would be potential. Hence, in my opinion, a complete re-writing and restructuring is needed. Several scientific terms are not appropriate. English must be corrected.

Some suggestions are:

-        Change the title to highlight the potential of this article and better show the objectives. Why « Genetic modification » ?

-        Change the title of the sections: sevral of them do not correspond to the section text and are not enough informative.

-        Introduction: add some information on pullulan production (yield, volumes), other reports on pullulan production enhancement , polysaccharide production enhancement techniques such as fermentation optimization, strain engineering including mutagenesis techniques.

-        Better present the BIFC method. What is the contribution of this work to this method ; is it new (Cf ref 8)? What are proteins a and b? Why does the case in the Figure 1A occur? and is it not a source of underestimation of the amount of pullulan? What is « fusion »? What is PULLCBM (you change the name here).

-        Section 2.7 Fermentation. Change the title because it deals with cultivation in microplates which is not fermentation. The removal of 20 µL (line 162) is not clear at all, what are they used for, are they seeded in a new liquid medium? Line 164: do you really use the whole fermentation broth (with whole cells) for pullulan assay?

-        Section 2.9: why is grinding necessary, isn’t there an effect on pullulan molecular weight?

-        Figure 2: we can see standard deviation. Statistical analysis should be performed to compare the results: maybe there is a decrease after 1h (and no stabilization as indicated on line 2012). What are the units (also for lines 212 and 214).

-        Section 3.2: how is calculated the mortality? Do 2 digits after the comma mean anything? What was the initial cell number?

-        Section 3.3: I’m lost. What is FITC? What does it measure? What is a « screening intensity of 1/10000 mean »? Line 242 has to be explained.

-        Figure 4 must be better explained and analysed: what are your conclusion?

-        Section 3.4. What is the number of mutants screened, only 6? All the results from the screening should be presented.

-        Line 259: I cannot see « 26.5 g/L » on the figure.

-        Results in Figure 6 should be discussed: relationship between viscosity, pullulan concentration and molecular weight.

-        Line 275: what is « actual production »

-        Lines 276-279: should be placed in the previous section.

-        Line 271: what is the volume of the medium in the 5L tank?

-        Discussion contains general information and do not discuss the study results in relation to bibliographic data. It mus t be rewritten.

-        Summary is not necessary, it is redundant with the conclusions. Conclusions do not correspond to the results: there is no indication that the detection technique can be used for other polysaccharides, no discussion on the proteins a and b, neither on the CBM. And lines 325-329 talk about limitations.

-        Supplemental data is not cited in the text.

Comments on the Quality of English Language

Several scientific terms are not appropriate. English must be corrected.

Reviewer 4 Report

Comments and Suggestions for Authors

The author's focused on increasing pullulan production in A. pullulans. The manuscript focuses on two spects: 1) Developing a high throughput screening assay for pullulan detection; 2) UV mutagenesis to increase pullulan titers. The manuscript has flaws in experimental design that does not support all the claims and leaves many unanswered questions to be considered a novel study to be considered for publication in its current state. The major comments are listed below:

a) The motivation behind this study is unclear from introduction and discussion section, considering there are plethora of publication available on pullulan production in A. pullulans with better titers.

b) The development of a screening method (Figure 1 and Figure 2) is questionable as the results does not justify the significant improvement in fluoresence that can be the indicative of very high pullulan titer.

c) There are some technical flaws to point out for developing screening assays related to execution. For example, appropriate controls (positive and negative) are not considered for developing this screening assay. PBS as negative control is not an appropriate control for this sort of assay development. There need to be inclusion of multiple controls to eliminate any sort of basal fluoresence including a benchmark strain as positive control with plus minus each probe (1a and 1b).

d) Figure legend's are not well defined so it is not comprehensible what conditions were tested.

e) Claims made regarding results seems over-exaggerated. For example Figure 2B and Figure 5, the author mentioned that fluoresence in test strain is significantly changed compared to control strain. Does 1.3 ~ fold change is substantial enough? Even basal fluoresence levels are extremely high in control strain which will make assay feasibility questionable in the events of high pullulan titer due to saturation in fluoresence signal using spectrophotomter.

f) The author's mentioned genetic modification in title. What genetic modification author's have performed to increase the titers? UV mutagenesis was performed but there is no knowledge of which mutations has increased the titers. If numerous isolates have been picked I am sure there will be even higher titer. 

g) Tracking of mutation that contribute to higher titer is important as that will add to its novelty. How one can replicate the results without any knowledge of genetic modifications?

h) The way manuscript is written there are lot of issues that makes it difficult for readers to follow up on material and methods. For example, in result section every section has very limited information in terms of execution and strain's and condition selection that it is not possible to understand why a particular condition was selected at first place.

i) Some sections in Discussion are not required. For example FCM technology instead the focus should be how current manuscript is addressing the roadblocks which previous manuscripts failed to address.

Comments on the Quality of English Language

The author needs to work on reorganization of their section. English language is fine but the way sections are organized is not establishing the motive behind any experiment. For example, in discussion section, instead of discussing results the authors are discussing about FCM technology in detail. Introduction, results, Material and methods are not written well so need modification.

Round 2

Reviewer 4 Report

Comments and Suggestions for Authors

I appreciate the efforts made by the authors to include the suggestions made by me in my previous review. It has improved the quality of manuscript compared to the previous version. However, I am still not convinced by the responses from authors in rebuttal letter. The latest version mostly focused on rearranging the sections to include more information from literature but the technical concerns raised in previous review still persist. 

I believe more experimental data should be included to track down the genetic modifications that actually contributed to increase in titer to increase the impact of this manuscript to overall contribute to novelty of this study. I have some concerns regarding the screening assay considering the basal expression is still very high and increase in signal is not enough to consider if it is a robust assay at high throughput scale. Therefore, all the claims can not be verified by current data.

Comments on the Quality of English Language

English language is fine with minor editing it can be accepted.